

# Some like it hot: factors impacting thermal preferences of two Ponto-Caspian amphipods *Dikerogammarus villosus* (Sovinsky, 1894) and *Dikerogammarus haemobaphes* (Eichwald, 1841)

Michał Rachalewski[1], Jarosław Kobak[2], Eliza Szczerkowska-Majchrzak[3] and Karolina Bącela-Spychalska[1]

[1] Department of Invertebrate Zoology and Hydrobiology, University of Lodz, Lodz, Poland
[2] Department of Invertebrate Zoology, Faculty of Biology and Environmental Protection, Nicolaus Copernicus University of Torun, Torun, Poland
[3] Department of Ecology and Vertebrate Zoology, University of Lodz, Lodz, Poland

Corresponding author
Michał Rachalewski,
rachalewski@me.com

## ABSTRACT

Temperature is a crucial factor determining biology and ecology of poikilothermic animals. It often constitutes an important barrier for invasive species originating from different climate zones but, on the other hand, may facilitate the invasion process of animals with wide thermal preferences and high resistance to extreme temperatures. In our experimental study, we investigated the thermal behaviour of two Ponto-Caspian amphipod crustaceans—*Dikerogammarus villosus* and *Dikerogammarus haemobaphes*. Both species are known to live under a wide range of thermal conditions which may promote their invasion. Moreover, both these amphipods are hosts for microsporidian parasites which co-evolved with them within the Ponto-Caspian region and spread in European waters. As the presence of a parasite may influence the thermal preferences of its host, we expected to observe behavioural changes in infected individuals of the studied amphipods leading to (1) behavioural fever (selecting a warmer habitat) or (2) anapyrexia (selecting a colder habitat). The experiment ($N = 20$) was carried out for 30 min in a 100 cm. 20 cm from boths sides were not avaliable for amphipods long thermal gradient (0–40 °C), using 30 randomly selected adult amphipod individuals of one species. At the end of each trial, we checked the position of amphipods along the gradient and determined their sex and infection status (uninfected or infected by one of microsporidium species). *D. villosus* was infected with *Cucumispora dikerogammari* whereas *D. haemobaphes* was a host for *C. dikerogammari*, *Dictyocoela muelleri* or *D. berillonum*. Thermal preferences of amphipods depended on their species and sex. Females of *D. villosus* preferred warmer microhabitats (often much above 30 °C) than conspecific males and females of *D. haemobaphes*, whereas no significant differences were found among males of both species and both sexes of *D. haemobaphes*. Moreover, infected males of *D. villosus* stayed in warmer water more often than uninfected males of this species, selecting temperatures higher than 30 °C, which may be explained either as a behavioural fever constituting a defence mechanism of a host against the infection, or as a parasite manipulation of the host behaviour increasing the parasite

fitness. On the other hand, none of the parasite species affected the thermal preferences of *D. haemobaphes*, including also *C. dikerogammari*, changing the behaviour of *D. villosus*. Our research presents the complexity of the thermal behaviour of studied amphipods and the evidence that microsporidia may trigger a change in temperature preferendum of their host species and those observations may be the result of different host-parasite coevolution time which may vary for the two host species (*Poulin, 2010*).

## INTRODUCTION

Temperature is the main variable determining multiple aspects of the functioning of poikilothermic organisms. It shapes their metabolic rate and thus their responses to other environmental factors, feeding, oxygen demands, growth rate and reproduction (*Liu & Walford, 1972*; *Newell, 1966*; *Cox & Rutherford, 2000*; *Sardiña et al., 2017*). In the aquatic environment, temperature can vary both vertically (due to stratification, usually defined as a change >1 °C m$^{-1}$) (*Wetzel, 2001*), and, particularly in shallow waters, horizontally (due to variable shading and solar radiation), shaping oxygen concentration, food availability and decay rates. Moreover, daily and seasonal changes in temperature take place in water bodies (e.g., *Irons III et al., 1994*; *Moore et al., 1997*; *Schindler, 1997*). This gives aquatic organisms the possibility of adjustment and optimization of their microhabitat selection with regard to temperature conditions by behavioural thermoregulation (*Macan, 1961*; *Cox & Rutherford, 2000*). Temperature can also be an important factor determining the probability of success of alien species, being the main barrier for those originating from different climate zones (*Gherardi, 2007*), but also facilitating invasions due to global changes taking place in the present world (*Van der Velde et al., 2002*; *Rahel & Olden, 2008*). Inability to match local environmental conditions, with a leading role of thermal regime, is considered as a major reason of failures of alien species establishment in novel areas (*Gherardi, 2007*). Therefore, knowledge of thermal preferences of alien species is crucial for understanding their invasive potential. Among invasive aquatic organisms, crustaceans belong to the richest groups in terms of species number (*Gherardi, 2007*) and their recent spread can be associated with global warming (*Stachowicz et al., 2002*; *Maazouzi et al., 2011*; *Hulme, 2017*).

A thermal range within which animals are most frequently observed is regarded as the preferred one (*Cossins & Bowler, 1987*). Temperature preferences can be influenced by the presence of parasites or pyrogenic substances (*Casterlin & Reynolds, 1977*; *Casterlin & Reynolds, 1979*; *Reynolds, Casterlin & Covert, 1980*). A behaviour called behavioural fever, consisting in the selection of higher temperatures compared to uninfected individuals, has been reported in many invertebrate taxa (*Elliot, Blanford & Thomas, 2002*; *Roy et al., 2006*; *Zbikowska & Cichy, 2012*). The function of such behaviour is pathogen suppression or elimination (*Roy et al., 2006*). In contrast to behavioural fever, the inverted fever or anapyrexia is characterised by the change in thermal preferences to colder microhabitats

(*Satinoff, 2011*) which may result in the decrease in parasite growth rate or proliferation time. This form of defence against pathogens has been extensively studied in snails by *Zbikowska (2004)*, *Zbikowska (2005)* and *Zbikowska (2011)* and *Zbikowska & Cichy (2012)*. Up to date, invertebrate anapyrexia is better recognized as a response to low oxygen concentration (*Morris, 2004*; *Gorr et al., 2010*), but its role in parasite-induced infections cannot be excluded. In the light of the constant arm race in parasite-host interactions, both host organisms and parasites developed strategies to increase their fitness. Whereas a host organism benefits from preventing a parasite from utilization of its resources, parasites influence the behaviour of their hosts to complete their life cycle and/or stimulate their spread in the host population. Parasites are an important component of biological communities and exert a strong impact on their structure and composition (*Hudson, Dobson & Lafferty, 2006*; *Lagrue, 2017*). Thus, numerous studies show that the presence of some species in the community may depend on the action of parasites, which modify their fitness, interactions with other community members and behaviour (*MacNeil et al., 2003*; *Dunn, 2009*).

Amphipods, are one of the most important components of freshwater ecosystems (*Piscart et al., 2009*), often used as model organisms in ecotoxicological (e.g., *Mehennaoui et al., 2016*), phylogeographical (e.g., *Grabowski et al., 2017*) and behavioural studies, including host-parasite interactions (*Rigaud, Perrot-Minnot & Brown, 2010*; *Petney, 2013*). Like other crustaceans, they are poikilothermic organisms and temperature is a crucial abiotic agent influencing their functioning (*Maranhão & Marques, 2003*). They rely on behavioural thermoregulation (*Lagerspetz & Vainio, 2006*). Although behavioural mechanisms of thermal preference of amphipods and the impact of parasites on their thermal behaviour have been scarcely studied, there are some examples where their thermoregulation has been evidenced (*Meadows & Ruagh, 1981*; *Timofeyev, Shatilina & Stom, 2001*). Nevertheless, temperature has been reviewed by *Sainte-Marie (1991)* as a key environmental variable that rules reproductive bionomics of Amphipoda and therefore constitutes a major agent shaping their life history traits. Hence, for instance a negative relationship between temperature and maximum size of individuals has been reported by *Panov & McQueen (1998)*. Additionally, it has also been shown that higher temperature results in producing more offspring, stimulates growth rate and influences amphipod activity (*Maranhão & Marques, 2003*; *Becker et al., 2016*). On the other hand, an interaction between high temperature and long day photoperiod may cause sex bias in a population by the emergence of intersexual individuals (*Dunn, McCabe & Adams, 1996*) and contribute to life cycle disturbances (*Neuparth, Costa & Costa, 2001*). Furthermore, the increase in temperature stimulates active brood care by females (*Dick, Faloon & Elwood, 1998*).

We considered Ponto-Caspian amphipod species as a perfect model for studies about temperature preferences due to their tolerance to relatively wide thermal regimes (*Pöckl, 2007*; *Bacela, Konopacka & Grabowski, 2009*; *Rewicz et al., 2014*), which is one of the features contributing to their invasion success. Furthermore, both within the invaded territories and their native range, Ponto-Caspian gammarids are infected with several sympatric microsporidian parasites (*Wattier et al., 2007*; *Ovcharenko et al., 2008*; *Ovcharenko et al., 2009*; *Wilkinson et al., 2011*; *Grabner et al., 2015*) which may influence their behaviour

(*Bacela-Spychalska, Rigaud & Wattier, 2014*) or induce sex ratio distortion (Green *Green Etxabe et al., 2015*). We have chosen two Ponto-Caspian invasive gammarid species: *Dikerogammarus villosus* (Sovinsky, 1894) and *Dikerogammarus haemobaphes* (Eichwald, 1841) as model organisms. They are ecologically similar and share evolutionary and invasive history as they often co-occur in their native and invaded ranges (e.g., *Dedju, 1980*; *Berezina, 2007*; *Grabowski, Jażdżewski & Konopacka, 2007*; *Leuven et al., 2009*). Both *Dikerogammarus* species can be infected by several microsporidian parasites: predominantly *Cucumispora* spp. and *Dictyocoela* spp. (*Wattier et al., 2007*; *Ovcharenko et al., 2010*; *Bacela-Spychalska et al., 2012*; *Bojko et al., 2015*; *Grabner et al., 2015*; *Green Etxabe et al., 2015*). These species often reach extremely high prevalence within the populations of their hosts (*Ovcharenko et al., 2010*; *Bacela-Spychalska et al., 2012*; *Bojko et al., 2015*; *Grabner et al., 2015*; *Green Etxabe et al., 2015*). They differ with regard to their infection pattern (vertical and/or horizontal transmission) and effect on their hosts (e.g., *Wattier et al., 2007*; *Krebes et al., 2010*; *Wilkinson et al., 2011*; *Bacela-Spychalska et al., 2012*; *Bacela-Spychalska, Rigauld & Wattier, 2014*; *Grabner, 2017*).

We hypothesized that (1) no differences regarding temperature preferences would be observed between amphipod species due to their common life history and similar environmental requirements. Thus, experimental protocol was designed to perform experiments separately for both species to exclude interspecific interactions among them (*Kobak, Rachalewski & Bacela-Spychalska, 2016*), which could give a false impression of thermal preferences. Furthermore, we assumed that (2) selection of the most favourable temperature by gammarids, regardless of species, could be conditioned by sex. This is due to the fact that females might be attracted to higher temperatures to hasten their maturation and oocyte development (*Panov & McQueen, 1998*) which would result in their higher fecundity (*Steele & Steele, 1973*; *Sheader, 1983*). Finally, we hypothesised that (3) the thermal preference of host species would be changed in the presence of a parasite, either due to the host defence mechanism (behavioural anapyrexia or fever), or due to the host manipulation by the parasite.

## MATERIALS & METHODS

### Animal collection

The two *Dikerogammarus* species were sampled from stable populations at two independent sites in the Vistula River (Poland) at a similar water temperature (c.a. 16 °C) in July. *Dikerogammarus villosus* individuals were captured from the Włocławek Reservoir (the lower River Vistula, Central Poland, N52.617738, E19.326453) and *Dikerogammarus haemobaphes* were collected from the middle part of Vistula River near Połaniec town (N50.423014, E21.311748). Population structure and species composition at both sites are well recognized by our research group and species determination was based on clear differences in body proportions, appendage setation (especially antennae II), as well as the arrangement and size of the conical projections of the urosome (*Eggers & Martens, 2001*; *Konopacka & Jazdzewski, 2002*). Animals were brought from the field in 3-L plastic buckets with aerated water which were placed in styrofoam boxes filled with ice packs.
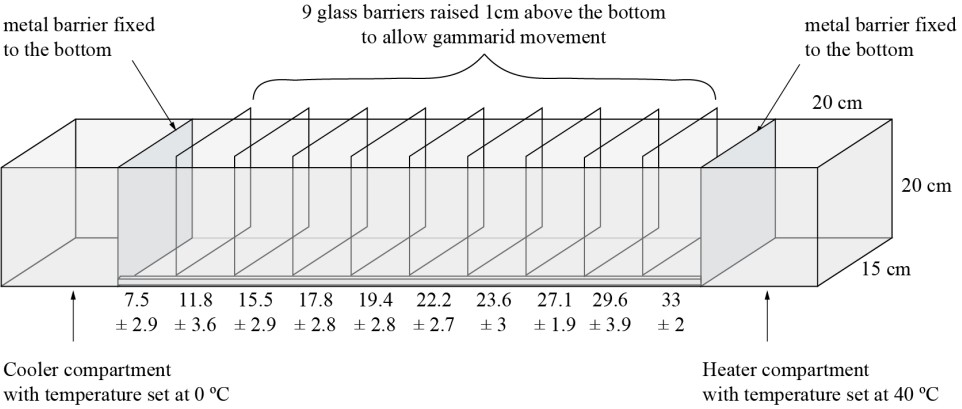

metal barrier fixed to the bottom

9 glass barriers raised 1cm above the bottom to allow gammarid movement

metal barrier fixed to the bottom

20 cm

20 cm

15 cm

| 7.5 | 11.8 | 15.5 | 17.8 | 19.4 | 22.2 | 23.6 | 27.1 | 29.6 | 33 |
| ± 2.9 | ± 3.6 | ± 2.9 | ± 2.8 | ± 2.8 | ± 2.7 | ± 3 | ± 1.9 | ± 3.9 | ± 2 |

Cooler compartment with temperature set at 0 °C

Heater compartment with temperature set at 40 °C

10 x 10-cm long compartments with a **thermal gradient** mean temperatures ± SD (°C) in particular compartments are presented

**Figure 1    Experimental tank and design.**

Prior to the experiments, animals were acclimatized in an air conditioned laboratory for one week at 20 °C, which is considered as an optimal metabolic temperature (*Bruijs et al., 2001*). Furthermore, we assessed the population structure, unveiling that the number of egg-carrying females was too low to include them as a separate group in statistical analyses. Knowing the life strategy of the both species (*Pöckl, 2007*; *Pöckl, 2009*; *Bacela, Konopacka & Grabowski, 2009*) we assume that in July all females showed high reproductive activity. Therefore, even if currently not carrying eggs, females were preparing for their release. Thus, all the females were pooled for the analysis as being likely to search for optimum conditions for oocyte/egg development.

Amphipods were fed daily with frozen chironomid larvae. After the acclimatization, the survival rate was high and animals did not exhibit any abnormal behaviour. We measured the lengths of 50 randomly selected animals under a stereoscopic microscope, from the base of the antennae to the tip of the urosome, with 0.1 mm accuracy. The mean body length was 13.2 mm (±3.6 mm SD) and 11.2 mm (±4.1 mm SD) for *D. villosus* and *D. haemobaphes*, respectively.

## Experimental setup

The experiment was performed in a 140 cm long, 15 cm wide and 20 cm high aquarium (Fig. 1) filled with aerated dechlorinated tap water to the water level of 10 cm. At the opposite sides of the aquarium, 20 cm from both end walls, we installed aluminum barriers (enabling good heat transduction). In one end of the tank we placed a heater set to 40 °C and in the other end—a water cooler (Teco R20; Teco S.l.r. Ravenna, Italy) set to 0 °C. The remaining space within the aquarium was divided with glass plates into 10 compartments (10 cm long each), leaving a 1.5 cm gap at the bottom to allow amphipods to migrate freely along the tank. These incomplete barriers helped maintain a thermal gradient. The

bottom of each compartment was equipped with a digital thermometer (accuracy up to 0.1 °C). We ran the system for 40 min before animals were introduced which was enough for the thermal gradient to be established. Then, we introduced 30 randomly selected adult amphipod individuals of one species to the compartment where the temperature was the same as that in which the animals were kept in the laboratory. Animals were left in the experimental aquarium for 30 min to migrate towards their preferred temperature. At the end, the glass barriers were inserted down to the bottom to prevent further relocation of the amphipods, which were collected from each compartment and placed in separate vials with 96% EtOH for sex identification and molecular screening of microsporidia. Preliminary observations have indicated that gammarids did not relocate much after initial selection of a thermal zone, thus the determination of their final distribution at the end of exposure produced similar results as their continuous observation. The experimental procedure was replicated 20 times both for *D. villosus* and *D. haemobaphes*. Thus, 600 individuals of each species were tested in the experiment and, to avoid pseudoreplication, each individual was used only once in this research. Nevertheless, for the statistics we selected only individuals infected with a single microsporidium species. Thus, thermal preferences of 600 individuals of *D. villosus* and 568 individuals of *D. haemobaphes* were finally analyzed. Amphipods are highly gregarious animals (*Sornom et al., 2012*; *Labaude, Rigaud & Cézilly, 2017*) and our experiment was designed taking this trait into account. That is why we decided to test them in groups. Thus, any interactions between the individuals in the experimental tank reflected their natural conditions. Nevertheless, the individuals after the experiment were not injured nor damaged, which suggested that antagonistic interactions were not common. In our study, we did not perform neither chemical nor physical analysis of water in the experimental tank. Preliminary tests and earlier experiments carried out in the same experimental setup (*Kobak et al., 2017*) showed that water quality (conductivity, pH, oxygen) was suitable for gammarids and did not change significantly during the trial. Oxygen concentration was not measured due to the risk of disturbance of the thermal gradient by handling the probe.

## Microsporidium detection and identification

Microsporidian DNA was co-extracted with host DNA with Chelex® sodium procedure by Sigma-Aldrich company following the protocol of *Casquet, Thebaud & Gillespie (2012)*. Afterwards, a PCR was conducted and a pair of microsporidia-specific primers V1f/530r (following *Baker et al., 1994* and *Vossbrinck et al., 1993*) was used to amplify distinctive parasite DNA fragments. PCR reactions were run in 10 µl reaction mixtures with each primer concentration of 400 nM, 200 µM dNTPs and 0.5 U/µl Promega Taq polymerase. The product was amplified under the following PCR conditions: an initial denaturing step at 95 °C for 2 min was followed by 35 cycles of 95 °C for 30 s, 62 °C for 45 s and 72 °C for 1 min. These cycles were followed by a final extension at 72 °C for 5 min. The PCR product was visualized on the 2% agarose gel in order to identify the infected individuals (positives). All the PCR products were purified with exonuclease I (Burlington, Canada) and FastAP alkaline phosphatase (Fermentas, Waltham, MA, USA) treatment and sequenced directly with the BigDye technology by Macrogen Inc., (Amsterdam,
The Netherlands) using the above mentioned primers. The obtained microsporidian sequences were edited using Geneious R10 (http://www.geneious.com, *Kearse et al., 2012*). Afterwards, identification of the microsporidia was determined using BLAST in GenBank (https://www.ncbi.nlm.nih.gov/genbank/).

## Data analysis

As both species were infected by different parasite species (see Results), we analysed their temperature selection separately, using a three-way General Linear Model with the following independent terms: (1) Infection (two levels: infected or not for *D. villosus*, three levels: two parasite species or uninfected for *D. haemobaphes*); (2) Sex; (3) Infection × sex interaction and (4) Replicate, a random factor included to account for the assignment of various individuals to particular aquaria and thus avoid pseudoreplications. We used temperatures of aquarium compartments in which particular amphipods were found as a dependent variable.

Moreover, to check for the differences between both species, we compared uninfected individuals with a 3-way GLM with (1) Species; (2) Sex; (3) Species × sex interaction and (4) Replicate (a random factor nested in Species) included in the model.

Post-hoc tests for significant effects were conducted using sequential-Bonferroni corrected Fisher's Least Significant Difference tests. Homoscedasticity and normality of data assumptions were checked with a Levene and Shapiro–Wilk tests, respectively. All the statistical tests were carried out with IBM® SPSS® 24.0 (Chicago, IL, USA).

# RESULTS

## Microsporidia species

Both *Dikerogammarus* species were infected with microsporidia; however the prevalence of microsporidiosis was higher in *D. villosus* (52%, $N = 600$) than in *D. haemobaphes* (27.4%; $N = 583$). The gammarid hosts differed in the parasite species composition: *D. villosus* was infected only with *Cucumispora dikerogammari*, whereas *D. haemobaphes* was infected by *C. dikerogammari*, *Dictyocoela berillonum* and *Dictyocoela muelleri*. Occasionally, there were mixed infections in *D. haemobaphes* by both *Dictyocoela* spp. and *C. dikerogammari* (13 individuals), and those cases were rejected from further analyses due to the risk of misinterpretation caused by ambiguous and diffuse influence of both parasites on their host. A detailed summary of microsporidiosis prevalence in the host species is presented in Table 1. Sex proportion of animals used in experiments and presented as females:males ratio was 1:0.7 for *D. villosus* and 1:0.9 for *D. haemobaphes*.

## Temperature preference of amphipods

The thermal behaviour of *D. villosus* (Figs. 2A–2D) depended on an interaction between sex and parasite presence (Table 2, Figs. 2A–2D). Post-hoc tests revealed that uninfected males preferred to stay at colder temperatures than other individuals, which mainly selected temperatures above 30 °C. On the other hand, uninfected males could be divided into two groups, one of them selecting low temperatures (<16 °C) and the other occupying high temperatures (>30 °C). We did not observe any significant effects of sex and parasite

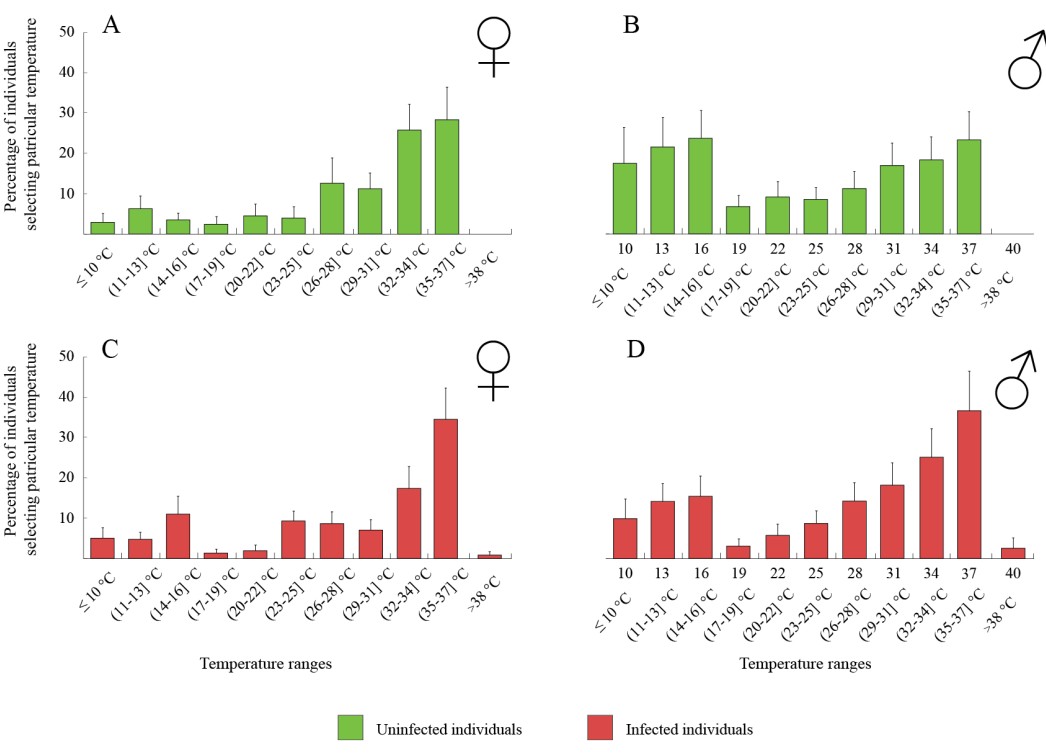

**Figure 2 Temperature selection of *Dikerogammarus villous* individuals in the thermal gradient with distinction of sex and infection presence.** (A) represents total percentage of uninfected females, (C) infected females, (B) uninfected males and (D) infected males. Error bars represent standard errors of the means.

**Table 1 Percentage of uninfected host amphipods and prevalence of microsporidiosis caused by the following microsporidium species.**

|  | *Dikerogammarus villosus* | | *Dikerogammarus haemobaphes* | |
|---|---|---|---|---|
|  | Males (N = 358) | Females (N = 243) | Males (N = 272) | Females (N = 296) |
| Uninfected | 51% | 44% | 79% | 72% |
| Infected with: |  |  |  |  |
| *Cucumispora dikerogammari* | 49% | 56% | 8% | 12% |
| *Dictyocoela muelleri* | – | – | 7% | 7% |
| *Ddictyocoela berillonum* | – | – | 6% | 9% |

presence or parasite species on the thermal preferences of *D. haemobaphes* (Table 2, Figs. 3A–3H), which selected a wide range of temperatures, generally similar or higher than their acclimation temperature (20–34 °C). Healthy females of *D. villosus* selected higher temperatures than female *D. haemobaphes*, whereas the preferences of uninfected males did not differ significantly between species (Figs. 2 and 3), resulting in a significant species × sex interaction in the analysis of thermal preferences of uninfected gammarids (Table 2).

**Table 2  General Linear Model to test the impact of species, sex and parasite infection on thermal behaviour of the studied gammarids.** Asterisks indicate statistically significant results.

| Analysis | Effect | df | MS | F | P |
|---|---|---|---|---|---|
| A. *D. villosus* | Sex (S) | 1 | 2038.1 | 26.81 | <0.0001* |
| | Infection (I) | 1 | 402.5 | 5.30 | 0.0220* |
| | Replicate | 19 | 299.5 | 3.94 | <0.0001* |
| | S × I | 1 | 995.3 | 13.09 | <0.0001* |
| | Error | 578 | 76.0 | | |
| B. *D. haemobaphes* | Sex (S) | 1 | 218.1 | 3.83 | 0.0508 |
| | Infection (I) | 3 | 52.8 | 0.93 | 0.4268 |
| | Replicate | 19 | 154.5 | 2.72 | 0.0001* |
| | S × I | 3 | 81.8 | 1.44 | 0.2310 |
| | Error | 543 | 56.9 | | |
| C. Healthy individuals of both species | Species (Sp) | 1 | 169.4 | 2.62 | 0.1062 |
| | Sex (S) | 1 | 2632.8 | 40.70 | 0.0000* |
| | Replicate | 19 | 203.6 | 3.15 | 0.0000* |
| | Sp × S | 1 | 575.9 | 8.90 | 0.0030* |
| | Error | 620 | 64.7 | | |

## DISCUSSION

We demonstrated that *Dikerogammarus villosus* and *D. haemobaphes* differed with regard to their thermal preferences. Females of *D. villosus* preferred warmer habitats than females of *D. haemobaphes*. In fact, females of *D. villosus* often occupied the warmest part of the gradient, at temperature of more than 32 °C, likely to cause their death after a longer exposure (*Wijnhoven, Van Riel & Van der Velde, 2003*). This might have resulted from a thermal shock experienced by animals entering a lethal temperature zone, making them unable to move. Nevertheless, as they were always introduced to the gradient at the location thermally matching their acclimation temperature, they entered this zone by active movement. Thus, they were not repelled by extremely high temperatures. We suggest that both species may exhibit different strategies of temperature selection. Is it likely that *D. villosus* selects the highest available temperature, as also confirmed by *Kobak et al. (2017)*, whereas *D. haemobaphes* avoids extreme temperatures. Under natural conditions, both strategies would possibly lead to a similar effect, i.e., occupation of warm microhabitats of suitable thermal conditions. This is because extreme temperatures, as those used in the warm part of our gradient, are missing in the environment where both species live. Nevertheless, under artificial laboratory conditions it was possible to detect interesting differences in the mechanisms underlying habitat selection by both species. Males of *D. villosus* preferred to stay in similar thermal conditions as *D. haemobaphes* (regardless of sex). Elevated temperature is especially beneficial for females because it accelerates growth and development rate and there is a positive correlation between female body size and fecundity (*Bacela, Konopacka & Grabowski, 2009*; *Pöckl, 2007*). Furthermore, during the reproductive period, large, receptive females are more attractive to males (*Dick & Elwood, 1990*; *Borowsky, 1991*; *Krång & Baden, 2004*). On the other hand, the higher

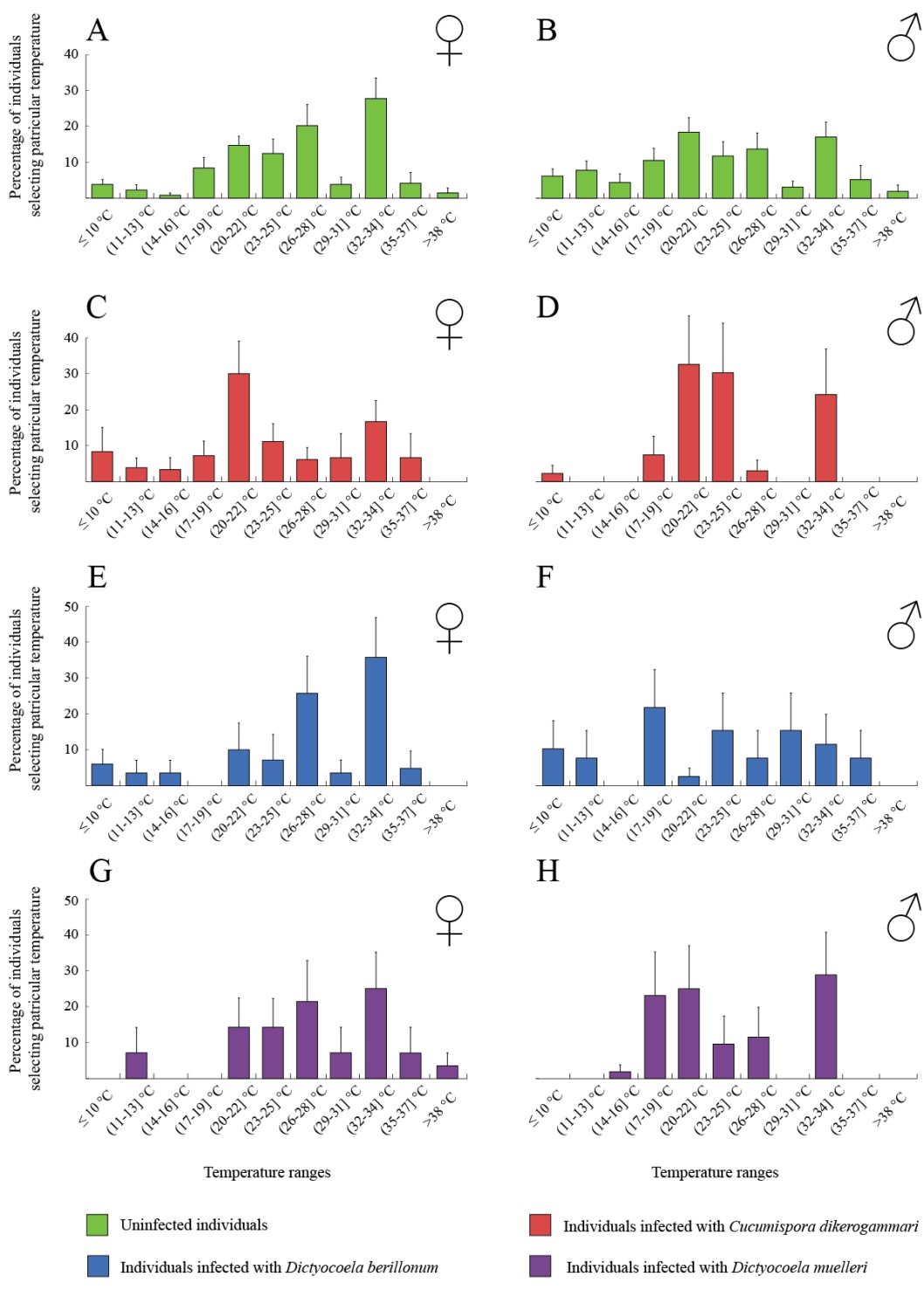

**Figure 3** **Temperature selection of *Dikerogammarus haemobaphes* individuals in the thermal gradient with distinction of sex and infection presence.** (A) represents total percentage of uninfected females, (C) females infected with *Cucumispora dikerogammari*, (E) females infected with *Dictyocoela berillonum*, (G) females infected with *Dictyocoela muelleri*, (B) uninfected males, (D) males infected with *Cucumispora dikerogammari*, (F) males infected with *Dictyocoela berillonum*, (H) males infected with *Dictyocoela muelleri*. Error bars represent standard errors of the means.

metabolic rate at higher temperatures results in the higher oxygen consumption and increase in the frequency of respiratory pleopod movements (*Dick, Faloon & Elwood, 1998*; *Wijnhoven, Van Riel & Van der Velde, 2003*; *Maazouzi et al., 2011*). At the same time, due to the lower solubility of oxygen in water, the probability of oxygen deficiencies increases with temperature. Thus, temperature-based habitat selection must be a trade-off between the acceleration of maturation and energy expenditure associated with higher metabolic demands. Oogenesis is a more energetically expensive process than spermatogenesis (*Bedulina et al., 2017*), thus the faster metabolism at the high temperature is likely to stimulate the former to a greater extent. Perhaps that is why most of the females selected high temperatures, whereas males exhibited two opposite thermal strategies, selecting either cold or warm water.

Due to the high impact of ecological disturbances caused by *D. villosus* in invaded territories, its ecology and biological limitations are well studied. Hence, *Wijnhoven, Van Riel & Van der Velde (2003)* reported that critical temperature threshold for *D. villosus* is estimated at 31 °C whereas *Maazouzi et al. (2011)* showed in a short time experiment that the temperature tolerance is lower: 26 °C. However, it has been demonstrated that *D. villosus* has reduced tolerance to higher temperature in waters of low conductivity (*Wijnhoven, Van Riel & Van der Velde, 2003*). Interestingly, *Hupalo et al. (2018)* showed that the temperature of 27 °C induces a thermal stress for *D. vilossus*, resulting in the production of heat shock proteins (HSP70). Nevertheless, *Hupalo et al. (2018)* proved that the resistance of *D. villosus* to thermal stress was different in two European genetically distinctive populations. This difference can be explained by different experiences of those populations, faced along two independent migration routes representing two different climatic-geographical regions. The western route is composed of the Danube, Rhine and the major French rivers, ending in the German river systems connected by an artificial channel with the Oder river in Poland. The eastern route is shorter and composed of the Dnieper, Prypiat, Bug and Vistula rivers. The eastern route is characterized with the harsh continental climate, whereas the western route is influenced by the calm Mediterranean and Atlantic climate. Hereby, *Hupalo et al. (2018)* found that animals migrating through the eastern route were less tolerant to high temperatures than those from the western route. However, although our experiments were performed on the population from the eastern route, individuals of *D. villosus* appeared to group at warm temperatures, often exceeding the stressful temperature of 27 °C (*Hupalo et al., 2018*). This supports our above-mentioned hypothesis on the strategy of choosing the highest available temperature, used by this species to select an optimum microhabitat. In contrast to *D. villosus*, *D. haemobaphes* tended to stay at intermediate temperatures in our gradient and seemed to be generally less selective with regard to temperature. Accurate data on temperature tolerance for this species are missing except those reported by *Kititsyna (1980)*, who stated that it was capable of surviving in a wide range of temperatures (6–30 °C). However, until now there have been no experimental data on the thermal preferendum of this species. "Both *Dikerogammarus* species preferred to stay at temperatures higher than that to which they were previously acclimatized. This is congruent with observations made on various crustacean species (*Kivivuori, 1994*; *Chen & Chen, 1998*; *Lagerspetz, 2000*; *Díaz et al., 2002*; *Lamkemeyer, Zeis & Paul, 2003*; *Kobak et*

*al., 2017*) whose thermal preferendum is also elevated in relation to an acclimatization temperature.

An important factor, which was expected to influence the thermal behaviour of amphipods was the presence of microsporidian parasites. Only in the case of *D. villosus* males were we able to detect the impact of microsporidian parasites on thermal behaviour: infected males tended to stay at higher temperature than uninfected ones. The observed temperature preferences could be an effect of either an action of the parasite or defensive response of the host to the infection. Hence, the explanation should be presented within such a dualistic scenario. This is because the sort of behavioural changes expected for a behavioural fever (a host act) may be similar to that expected if the parasite mediates the host behaviour for its own needs (*Poulin, 1995*). Behavioural changes triggered by parasitism in amphipods have been widely studied (e.g., *Bethel & Holmes, 1973*; *Cezilly, Gregoire & Bertin, 2000*; *Lagrue, Kaldonski & Perrot-Minnot, 2007*; *Lefèvre et al., 2009*; *Bakker, Frommen & Thünken, 2017*; *Labaude, Rigaud & Cézilly, 2017*). Parasites which are transmitted to another host trophically (e.g., acanthocephalans) are particularly considered as inducing behavioural modifications of their hosts to increase their susceptibility of being preyed by another, final host (*Holmes & Bethel, 1972*; *Lafferty, 1999*). These modifications include changes in the preference to light, staying outside shelter and entering the water column (*Cezilly, Gregoire & Bertin, 2000*; *Perrot-Minnot et al., 2012*). In our study, only the presence of microsporidium *Cucumispora dikerogammari* in *D. villosus* significantly modified amphipod behaviour. This parasite species relies mainly on trophic transmission and its development takes place mainly in its host muscles (*Ovcharenko et al., 2009*; *Bacela-Spychalska et al., 2012*). Therefore, its presence may lead to locomotion inability of amphipods (*Fielding et al., 2005*). However, contrary to this observation, *Bacela-Spychalska, Rigaud & Wattier (2014)* found that amphipods were more active when infected by this parasite. Muscle necrosis is a result of the fast spore development in muscle tissue and consequently can be beneficial for the parasite transmission as the host becomes more vulnerable to predation by cannibalistic conspecifics. We suggest that analogous situation could be observed in our research, where infected individuals of *D. villosus* aggregated in warmer water where metabolic expenditure increases, which in the wild would likely lead to the decrease in their survival time. Hence, the higher mortality of host individuals may in turn increase the possibility of microsporidia to transfer to another host, as *D. villosus* often expresses scavenging behaviour (*Dick, Platvoet & Kelly, 2002*).

Another explanation for the preference of infected *D. villosus* males towards warmer water may be the host defence response, which seems to better explain the observed behaviour. It is known that microsporidium virulence and survival is limited in both high and low temperatures, but the value of this temperature may vary in different microsporidian species (*Maddox, 1973*). Although extensive studies of the temperature thresholds affecting those parasites in amphipod hosts are missing, there are however examples of such studies in other invertebrates (*Raun, 1961*; *Olsen & Hoy, 2002*; *Martín-Hernández et al., 2009*). *Undeen, Johnson & Becnel (1993)* estimated the thermal tolerance of a microsporidium *Edhazardia aedis* infecting yellow fever mosquito (*Aedes aegypti*) as ranging from 0–40 °C. On the other hand, (*Olsen & Hoy, 2002*) showed that a temperature

of 33 °C in the aquatic environment is sufficient to cure the midge *Metaseiulus occidentalis* infected by microsporidia. *Weiss & Becnel (2014)* summarised on the basis of numerous studies that 35 °C is enough to severely reduce viability of microsporidian spores. As these are the values selected by infected *D. villosus* males in our study, it is possible that their observed affinity to higher temperature is a defence response to infection. Although animals present in the warmest compartments were less vital, it is possible that we observed a trade-off between a thermal stress for the host and possibility to decrease the viability of the parasite. Up to now, behavioural fever induced by microsporidia has been found in few invertebrates (*Boorstein & Ewald, 1987*; *Campbell et al., 2010*). The thermal behaviour of *D. villosus* presented in this study could likely be considered as new evidence for behavioural fever in invertebrates. Interestingly, we did not observe any changes in the thermal behaviour of females of *D. villosusd*. As we have demonstrated above, uninfected females of this species also selected warm water, so further increase in selected temperature by infected females was not possible.

None of the microsporidium species detected in *D. haemobaphes* affected thermal preferences of their host, including also *Cucumispora dikerogammari*, previously shown to change the behaviour of male *D. villosus*. The difference in responses between the gammarid species can be accounted for by their different habitat selection strategies, as suggested earlier, with *D. villosus* choosing the warmest available temperature and being generally more thermophilic than *D. haemobaphes*, which tends to avoid extremes. It may also be explained by the different host-parasite coevolution time which may vary for the two host species (*Poulin, 2010*).

Two other microsporidia infecting *D. haemobaphes*: *Dictyocoela berillonum* and *Dictyocoela muelleri* disseminate vertically (parental transmission to offspring), transovarially across maternal line (*Haine et al., 2004*; *Terry et al., 2004*). Compared to *C. dikerogammari*, *Dictyocoela* spp. are supposed to be less virulent, as their transmission depends on the reproductiveion success of their host (*Fine, 1975*; *Dunn & Smith, 2001*). They may lead to sex ratio distortion of *D. haemobaphes* population by feminizing the individuals (*Terry et al., 2004*; *Green Etxabe et al., 2015*), which however has not been found in our study. The lack of any action of the infected host to eliminate the microsporidium could be explained by its costs.

## CONCLUSIONS

In conclusion, we observed that *D. villosus* expressed a more complex thermal behaviour compared to *D. haemobaphes*. Both sex and microsporidian infection played an important role in the distribution of this species within a thermal gradient. Furthermore, our study showed that *D. villosus* was a more thermophilic species than *D. haemobaphes*. Using a temperature gradient we were able to unveil probable differences in temperature selection strategies between the tested amphipod species. We suggest different temperature preferences may contribute to the spatial segregation of these species and in consequence lead to the reduction in interspecific competition between them. The tendency to select high temperatures is likely to be especially beneficial when facing progressive climatic changes

(*Gallardo & Aldridge, 2013*; *Kernan, 2015*). Furthermore, *D. haemobaphes*, in contrast to *D. villosus*, is vulnerable to infection by more than one microsporidium species but, despite this, its thermal preferences are not affected by those parasites.

Knowledge about abiotic limitations of microsporidium species occurring in communities of invasive amphipods is particularly important for the conservation biology and for understanding the links between invasion patterns and disease transmission. This in consequence might help predict the impact of invaders on local biota (*Prenter et al., 2004*). It has been shown that the transmission of spores infecting Ponto-Caspian amphipods to native species is possible (*Bacela-Spychalska et al., 2012*, A Quiles, pers. comm., 2018).

Our study helps broaden the knowledge upon thermal preferences of two highly invasive species in Europe, taking into account an extremely important, inseparable factor affecting those animals—the presence of microsporidium parasites. Our results emphasize that the analysis of animal behaviour should be performed in such a wide perspective, especially because microsporidia are permanently associated with the invasive amphipod community in invaded territories.

## ACKNOWLEDGEMENTS

The authors wish to acknowledge Carola Winkelmann and Douglas Glazier for their substantial comments and linguistic consultations which allowed us to improve the quality of this manuscript.

### Funding

Our study was supported by the Polish National Science Centre grants 2011/03/D/NZ8/03012 (received by Karolina Bącela-Spychalska) and 2012/05/B/NZ8/00479 (received by Jarosław Kobak). The funders had no role in study design, data collection and analysis, decision to publish, or preparation of the manuscript.

### Grant Disclosures

The following grant information was disclosed by the authors:
Polish National Science Centre: 2011/03/D/NZ8/03012, 2012/05/B/NZ8/00479.

### Competing Interests

The authors declare there are no competing interests.

### Author Contributions

- Michał Rachalewski conceived and designed the experiments, performed the experiments, contributed reagents/materials/analysis tools, prepared figures and/or tables, authored or reviewed drafts of the paper, approved the final draft, sequence editing and analysing.
- Jarosław Kobak conceived and designed the experiments, analyzed the data, contributed reagents/materials/analysis tools, prepared figures and/or tables, authored or reviewed drafts of the paper, approved the final draft.

- Eliza Szczerkowska-Majchrzak authored or reviewed drafts of the paper, approved the final draft, laboratory work.
- Karolina Bącela-Spychalska contributed reagents/materials/analysis tools, authored or reviewed drafts of the paper, approved the final draft, sequence editing and analysing.

## Data Availability

The raw data are provided in a Supplemental Information 1.

## Supplemental Information

Supplemental information for this article can be found online at http://dx.doi.org/10.7717/peerj.4871#supplemental-information.

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
