# Peer review of "Some like it hot: factors impacting thermal preferences of two Ponto-Caspian amphipods Dikerogammarus villosus (Sovinsky, 1894) and Dikerogammarus haemobaphes (Eichwald, 1841)"

_PeerJ, doi:10.7717/peerj.4871_

## Round 0.1 · original submission · Major Revisions

Dear Authors,

The reviews of your manuscript have now come in and both reviewers see merit in your paper. One reviewer have highlighted that it is unclear whether the conclusions are supported by the data and have recommended that more detail is required in the methods/results to assess this fully. Please could you address the reviewers concerns and resubmit the manuscript.

·

Basic reporting

The paper of Rachalewski is mostly written in a clear and unambiguous English. Slight linguistic changes might be necessary. The introduction shows the background and context of the study well, although I do not concur with all statements (see general comments). The literature is well referenced and relevant. The structure of the paper conforms largely to PeerJ standards although some information are missing (see general comments). All figures are relevant and of good quality. I would like more information in the figure description (such as number of experimental repetitions, number of infected/uninfected animals). The raw data are partly supplied. I’m not exactly sure how to read the data file. Is the ID in the column “Aquaria” the ID for the specific experiment? I feel that the values for temperatures in the single partitions of the flume are lacking.

Experimental design

The research presented here is original primary research well within the scope of the journal. The research questions (in that case 3 hypotheses) are well defined and meaningful. The questions are relevant and it is clearly stated in the introduction how they will fill a knowledge gap. However, I do not see, how the hypothesis 3 and 4 tested with the data set presented here. It is only possible to describe the temperature preferences and possibly their changes due to parasites. It will however not be possible to discriminate between behavioural fever and parasite induced behavioural changes because the observed shift in the thermal preferences would be the same and no mechanistic analysis has been performed.
The experimental design, using temperature gradients in flumes during multiple trials is sound and the number of replication (n=20) seems adequate. The fact that the experiments contain individuals of different sex and infection status might represent an experimental problem because the individuals of the different groups might interact with one another. However, I do not see how this problem might have been avoided, because the sex-determination of living animals represents quite a lot stress for the animals and I belief a sound determination of infection status is not possible at all. However, the problem should probably be addressed in the discussion.
The description of the methods is not sufficient to evaluate the data and to replicate the experiments in its present form. I belief following information are missing: time of sampling, accuracy of species determination on living animals, reproduction status of females (carrying eggs?), temperature of experimental compartments (and its variation), kind of water used for keeping and in the experiments, chemical and physical conditions during the experiments (oxygen, conductivity, ammonia concentration), how was the sampling accomplished in the flume without allowing the animals to change their position. I expect that each individual was only used once. However there is no mention of that fact in the methods. In addition, there is no information regarding the statistical software used.

Validity of the findings

Although the experiment might conducted correctly, which I cannot assess because of the missing information, I feel that the statistical analysis is not adequate. The authors use the temperature at which every single animal was observed after 30 min as the response variable. The aim of the statistical test is however to find a potential shift of the temperature preferendum. Consequently, the temperature preferendum should be used as the response variable. In addition, the method used by the authors handles every single individual as an independent n, which is not true and uses the respective trial as random factor. In my opinion, the trials are real and independent replicates and the animals within each trial pseudo-replicates. And as the temperature preferendum is addressed in this paper, I recommend that the authors use the median or mean temperature of each trial as response variable (see for instance Reiser et al 2014, http://dx.doi.org/10.1016/j.jembe.2014.08.018; or Haupt et al in press http://dx.doi.org/10.1016/j.jinsphys.2016.12.006 ). In addition, such behavioural experiments often produce artefacts because individuals become caught at the temperature extreme due to physiological stress and consequent loss of mobility. Because the distribution in case of D. villosus is skewed towards the upper end of the temperature gradient, this aspect should be at least be addressed in the discussion.
I cannot assess, whether the conclusions are supported by the data provided here due to the above mentioned statistical issues. This would have to be assessed after a revision of the manuscript.

Additional comments

As mentioned above, the paper is well written, focussed on a clearly specified and relevant research question and therefore well within the scope of PeerJ. However, I feel that there are a lot of information missing to evaluate the method of the study and the validity of the data. In addition, I see major problems with the statistical analysis used here (see Validity of findings). Especially the right-skewed distribution of the relative abundance might be an experimental artefact unless there is in fact an 11th compartment that was hardly used by the animals (>38°C). However, as only 10 compartments are mentioned, I wonder why there are 11 temperature classes in the figures.
Further, I see some problems in the introduction. The authors state, that amphipods are a good model to study temperature preferences (L89 and 106) but do not support the statement sufficiently. It is not clear to me, why amphipods in general and Ponto-Caspian amphipods specifically are such a good model. In addition within the introduction, Rachalewski et al point out that temperature preference is an important species trait regarding invasional success (L108 f). However, firstly they do not support their statement with valid citations (containing real data and not only general assumptions) and secondly it is not really the point of the paper.
I do not comment the discussion here because I assume that after reanalysing the data, some statements might have to be changed. Generally, the discussion seems not to be completely focussed on the research question and might contain some slight over-interpretation of the patterns observed here, especially in the light of potential experimental artefacts.

·

Basic reporting

Specific comments:

L 36: To avoid confusion about the species of the males, I suggest reversing the order of this phrase “females of D. haemobaphes and conspecific males” to “conspecific males and females of D. haemobaphes”.

L 64, 90: Omit “the”.

L 67: Insert “the” before “preferred”.

L 106: Change “upon” to “about”.

L 107: Change “resistance” to “tolerance”.

L 109: Change “What is more” to “Furthermore”? Insert “their” before “native”.

L 126-146: I am a bit puzzled by the placement of parenthetical numbers in this paragraph.

L 158: Change “to the temperature of 20°C” to at “at 20°C”.

L 179: Change “to” to “and placed in”.

L 180-181: Somewhat awkward wording. Change “to identify their sex and proceed the molecular screening for the microsporidium presence and identification” to “for sex identification and molecular screening of microsporidia”?

L 217-219: Run-on sentence. Replace comma with semicolon between “microsporidea” and “however”.

L 220: Replace “while” with “whereas”. The word “while” has an unintended temporal connotation. Insert “was infected” before “by”.

L 249: Please clarify what is meant by “a similar effect”. What kind of similar effect?

L 256: Change “What is more” to “Furthermore”?

L 257: Change “large, receptive females are assessed by males as more attractive” to “large, receptive females are more attractive to males”.

L 258-260: Better references would be:

Wijnhoven et al. (2003). Exotic and indigenous freshwater gammarid species: physiological tolerance to water temperature in relation to ionic content of the water. Aquatic Ecology 37, 151-158.

Maazouzi et al. (2011). Ecophysiological responses to temperature of the “killer shrimp” Dikerogammarus villosus: is the invader really stronger than the native Gammarus pulex?. Comparative Biochemistry and Physiology Part A: Molecular & Integrative Physiology 159, 268-274.

These studies actually show how respiratory pleopod beating increases with increasing temperature in one of the species studied by the authors.

L 272: The word “proved” seems too strong. Are you 100% certain?

L 276: Which authors?

L 285: Change “though” to “although”?

L 293: Change “upon” to “on”.

L 300: Change “we were” to “were we”.

L 302: Change “at the higher temperature” to “at higher temperatures”.

L 310: Change “by trophic way” to “trophically”?

L 320: Change “in consequence” to “consequently”.

L 336: Change “to a range of 0 – 40oC” to “as ranging from 0 – 40oC”

L 344-345: Make more concise. Change “only in very few studies upon invertebrates of different taxonomic classification” to “in few invertebrates”.

L 358: Insert “by” between “for” and “their”.

L 364: Change “reproduction” to “reproductive”.

L 369-371: If D. villosus is more effective at preventing or eliminating infection by microsporidia, then why does this species show higher percent infection than does D. haemobaphes (as shown in Table 1).

L 374: “we may assume”? Did they show more complex thermal behavior or not? Isn’t this an observation, rather than an assumption?

L 378: Omit “applied”?

L 379: What is “it” referring to in this sentence?

L 381: Change “among” to “between”. Change “while” to “when”?

L 383, 387: Omit “the”.

L 386, 392: Change “The knowledge upon” to “Knowledge about”.

L 386-387: Change “the community” to “communities”.

L 393: Insert “an” before “extremely”.

Figures 2 & 3: What are the error bars?

Experimental design

Good.

Validity of the findings

Good.

Additional comments

This paper presents useful data on how sex and microsporidian infection affect the thermal preferences of two species of invasive amphipods. Overall, the paper is well organized and presented. The data analysis also seems sound. I have only relatively minor comments, given above under Basic Reporting.

---

## Round 0.2 · accepted · Accept

Dear Authors,

Thank you for making those changes to the manuscript. I'm delighted to now accept your manuscript for publication.

Kind regards,

Alex

#